# Separate Observer-based Estimation and Control for Unmanned Autonomous Vehicles with Disturbances and Faults under Input Saturation

1st Qianqian Zhang
*Department of Radiology*
*Yantai Yuhuangding Hospital of Qingdao University*
Yantai, China
wfcyzhangqian@126.com

2nd Jie Gao
*Department of Hepatobiliary & Pancreatic & Spleen Surgery*
*Yantai Yuhuangding Hospital of Qingdao University*
Yantai, China
29960308@qq.com

3rd Guojie Han*
*School of Hydraulic Engineering*
*Ludong University*
Yantai, China
hanguojie61@163.com

4th Xin Hu
*School of Mathematics and Statistics Science*
*Ludong University*
Yantai, China
931801933@qq.com

*Abstract*—This paper discusses the separate observer-based estimation and control for unmanned autonomous vehicles with disturbances and faults under input saturation effects. A trajectory tracking control scheme is designed with integrating the separate observers. The disturbance observer on-line estimates and compensates for time-varying uncertain disturbances. The actuator fault observer enables on-line estimation and compensation of actuator faults. An auxiliary dynamic filter is used to attenuate the adverse effects of control saturation. The Lyapunov stability theory demonstrates that all the signals of the closed-loop system are global uniformly ultimately bounded. Illustrations show the effectiveness of this scheme.

*Index Terms*—Unmanned Vehicle, tracking control, disturbance observer, fault observer, input saturation.

## I. Introduction

In recent years, the unmanned autonomous vehicles (UAVs) have been able to accomplish specific tasks without being operated and are widely used in fields such as oil and gas exploration, offshore monitoring, and seabed mapping [1], [2]. During the sea voyage, the UAV will not only be suffering from external ocean environment disturbances such as wind, waves and currents, but also be affected by the sudden faults of external actuators, thus reducing the stability and safety of the UAV motion control systems. Meanwhile, the physical limitations of the operating performance and UAV actuator make the command control signals bounded by the input saturation, which brings another challenge to the UAV's trajectory tracking control.

*Corresponding author. Guojie Han (hanguojie61@163.com). This research was funded partly by National Natural Science Foundation of China under Grant 62273172, partly by Outstanding Youth Innovation Team Project of Shandong Higher Education Institution under Grant 2021KJ042, and partly by Postgraduate Education Reform Project of Shandong Province under Grant SDYJG21045.

Due to the nonlinear nature of the UAV's equations of motion, the tracking control problem was first solved using nonlinear vector backstepping in [3]. The work of [4] proposed the design of ship trajectory tracking based on nonlinear backstepping method and adaptive backstepping method. The work of [5] proposed a trajectory tracking control method for fully actuated ships based on the idea of feedforward approximation. The work of [6] designed an under-actuated ship trajectory tracking control law using coordinate transformation in combination with backstepping. Considering that the UAV are subject to external time-varying disturbances, the work of [7] designed an adaptive disturbance observer to estimate the external environmental disturbance and applied the backstepping scheme to design the UAV trajectory tracking control law. The work of [8] used an observer to estimate the ship's velocity vector and design an observer-based trajectory tracking output feedback control law based on the velocity estimates. The work of [9] proposed an adaptive neural network tracking control scheme for ships under uncertain disturbances using a dynamic surface method. The work of [10] used neural network (NN) and dynamic surface techniques to design adaptive robust control laws for under-actuated UAV trajectory tracking. The work of [11] designed a unique fixed-time output feedback sliding mode tracking control method to solve the trajectory tracking problem of marine surface vessels (MSVs). The work of [12] proposed an adaptive output feedback control method based on a neural network for UAV trajectory tracking. Furthermore, various effective methods such as PID control, fuzzy control, model predictive control, sliding mode variable structure control, deep reinforcement learning, and neural networks have been proposed for research on UAV trajectory tracking control.

The aforementioned research results on the UAV trajectory

tracking control do not address the case of actuator fault. In the case of UAV actuator fault, the work of [13] proposes a dynamic surface control method based on a fault state observer to estimate the unknown fault state by designing a fault state observer. The work of [14] proposes a new fault tolerant control (FTC) method that forces the UAV to reach the desired position despite a thruster fault in a finite time. In fact, a single compensating control for UAV actuator fault is unreliable when the ship is disturbed. Thus, achieving trajectory tracking control of UAV in the presence of simultaneous external disturbances and actuator faults is a big challenge. For this reason, a finite-time observer-based actuator fault-tolerant control method was established in [15]. In addition, there are always bounds on the ship actuators in actual navigation, such as input saturation bounds. Once these physical boundaries are ignored in the system design, it may lead to weak closed-loop performance or even to system instability [16]. Therefore, reducing the effect of actuator saturation will benefit the stability performance of the control system. The work of [17] proposes an disturbance estimations method under actuator saturation conditions. The work of [18] combined disturbance observer and auxiliary dynamic system with dynamic surface control method to solve the problem of adaptive asymmetric adjustment control for dynamic positioning of ships with dynamic uncertainties. The work of [19] uses dynamic surface control techniques that weaken the actuator saturation effect while avoiding computational explosion. In addition, it is theoretically and practically important to research the trajectory tracking control of UAVs using independent observers in the case of saturated actuator inputs.

Inspired by the above discussed results, this paper considers the motion characteristics of the UAV under the influence of actuator input saturation and designs a UAV trajectory tracking controller with a disturbance observer and a fault observer. First, a disturbance observer is constructed for on-line estimation and compensation of unknown disturbances; second, a fault observer is constructed for on-line estimation and compensation of unknown faults; and third, an auxiliary dynamic filter is constructed for minimizing the detrimental effects of input saturation due to UAV actuators. Finally, simulation is performed to verify that the designed UAV trajectory tracking controller can achieve high control accuracy.

## II. PROBLEM FORMULATION

The two coordinate systems for establishing the USV are used consisting of the north-east frame and the body-fixed frame. We model USV's kinematics and kinetics [18] considering only three degrees of freedom at water surface as follows

$$\dot{\eta} = J(\psi)\nu \tag{1}$$

$$M\dot{\nu} + C(t)\nu + D(t)\nu = u(t) + d(t) \tag{2}$$

where $M$ is the inertia matrix, $C$ generally represents the centripetal force or even the Coriolis force matrix, and $D$ generally represents the damping term, $u(t) = [u_1(t), u_2(t), u_3(t)]^{\mathrm{T}}$ denoted the control vector for the USV. $u_1(t)$ is generally the force of the longitudinal swing change i.e. the change in the

direction of $CX$, $u_2(t)$ denoted the transverse swing force generated by the moving USV, which means the change in the direction of $CY$ and $u_3(t)$ denoted the change in the direction of the bow sway force $CZ$. $d(t) = [d_1(t), d_2(t), d_3(t)]^{\mathrm{T}}$ is denoted as the external external disturbances acting on the USV is subjected. $J(\psi) \in \mathbb{R}^{3\times3}$ represents the rotation matrix

$$J(\psi) = \begin{bmatrix} \cos(\psi) & -\sin(\psi) & 0 \\ \sin(\psi) & \cos(\psi) & 0 \\ 0 & 0 & 1 \end{bmatrix} \tag{3}$$

and has $J^{-1}(\psi) = J^{\mathrm{T}}(\psi)$.

### A. External Disturbances and Actuator Faults under Input Saturation

Due to the susceptibility of the command control signals of the USV internal actuators to the adverse effects of saturation, i.e., $u = Sat(u_c) = [sat(u_{c1}), sat(u_{c2}), sat(u_{c3})]^{\mathrm{T}}$ with the $sat(u_{ci})(i = 1, 2, 3)$ is the saturation function [20]

$$\mathrm{Sat}(u_{ci}) = \begin{cases} \mathrm{sign}(u_{ci})\tau_{Mi}, |u_{ci}| \geq u_{Mi} \\ u_{ci}, \qquad\quad |u_{ci}| \leq u_{Mi} \end{cases} \tag{4}$$

with $u_{Mi} > 0$, $i = 1, 2, 3$ being equivalent actuator saturation limits in surge, sway, and yaw, respectively, and $u_c = [u_{c1}, u_{c2}, u_{c3}]^{\mathrm{T}}$ being the commanded control vector calculated by the controller. Notate $\Delta u = Sat(u_c) - u_c$ as the control derivation.

The external time-varying disturbances $d(t)$ can be expressed as follows [21]:

$$\dot{d}(t) = -T^{-1}d(t) + \Psi n(t) \tag{5}$$

where $T \in R^{3\times3}$ represents the positive-definite time constant matrix. $\Psi \in R^{3\times3}$ is the Gaussian white noise amplitude matrix.

*Assumption 1:* The $n(t)$ denotes the bounded Gaussian white noise vector, $||n(t)|| \leq c_d < \infty$ and $c_d$ are also bounded unknowns.

We model the actuator dynamics as follows [22]:

$$\dot{u}(t) = -A_{tr}u(t) + A_{tr}u_c(t) + \omega(t) \tag{6}$$

Herein, $u_c(t) = [u_{c_1}(t), u_{c_2}(t), u_{c_3}(t)]^{\mathrm{T}}$ denotes the commanded thrust and moment vectors, and the matrix $A_{tr}$ is defined as

$$A_{tr} = diag\left(\frac{1}{T_{tr1}}, \frac{1}{T_{tr2}}, \frac{1}{T_{tr3}}\right) \tag{7}$$

where, the $T_{tr1}$, $T_{tr2}$ and $T_{tr3}$ denotes a time constant, and $\omega(t) = [\omega_1(t), \omega_2(t), \omega_3(t)]^{\mathrm{T}}$ denotes a time-varying fault that occurs in the USV internal actuator.

*Assumption 2:* The fault $\omega(t)$ generated by the USV's actuator is bounded.

*Remark 1:* The external environment and actuator fault are unpredictable and constantly changing. Unknown disturbances and faults acting on the USV are bounded due to the limited energy. Therefore, Assumptions 1-2 is reasonable.

## III. MAIN RESULTS

### A. Disturbance Observer Design

Design of disturbance observer for estimating and compensating the time-varying disturbances acting on the USV [23].

$$\hat{d}(t) = \phi + K_d \nu \tag{8}$$

$$\dot{\phi}(t) = -(T^{-1} + K)(\phi + K_d \nu)$$
$$+ K[C(\nu)\nu + D(\nu)\nu - u(t)] \tag{9}$$

where $\hat{d}$ is the estimate of the external disturbance and $\phi \in \mathbb{R}^3$ is the auxiliary middle vector generated by (9). $K_d \in \mathbb{R}^{3\times 3}$ is design parameter matrices which satisfy $K = K_d M^{-1}$.

Define the disturbance estimate error is

$$\tilde{d}(t) = d(t) - \hat{d}(t) \tag{10}$$

On the basis of (3), (8) and (9), we can obtain

$$\dot{\hat{d}}(t) = \dot{\phi} + K_d \dot{\nu}$$
$$= -(T^{-1} + K)\hat{d}(t) + K d(t) \tag{11}$$

From (11), we can obtain

$$\dot{\tilde{d}}(t) = -(T^{-1} + K)\tilde{d}(t) + \Psi n(t) \tag{12}$$

Select the Lyapunov function candidate as follows

$$V_d = \frac{1}{2}\tilde{d}^{\mathrm{T}} P \tilde{d} \tag{13}$$

Give a arbitrarily symmetric matrix $P$ (positive definite), there exists a symmetric matrix $Q$ (also positive definite) satisfying

$$-(T^{-1} + K)^{\mathrm{T}} P - P(T^{-1} + K) = -Q \tag{14}$$

According to the derivation of Lyapunov function (12)-(14), Young's inequality, it has

$$\dot{V}_d = \frac{1}{2}\big[ -(T^{-1} + K)\tilde{d} + \Psi n \big]^{\mathrm{T}} P \tilde{d}$$
$$+ \frac{1}{2}\tilde{d}^{\mathrm{T}} P \big[ -(T^{-1} + K)\tilde{d} + \Psi n \big]$$
$$= -\frac{1}{2}\tilde{d}^{\mathrm{T}} Q \tilde{d} + \frac{1}{2}\tilde{d}^{\mathrm{T}}\tilde{d} + \frac{1}{2}\|P\Psi n\|^2$$
$$\leq -\Big[\frac{1}{2}\lambda_{\min}(Q) - \frac{1}{2}\Big]\|\tilde{d}\|^2 + \frac{1}{2}\|P\Psi\|^2 c_d^2 \tag{15}$$

### B. Fault Observer Design

Building fault observers for estimating and compensating for time-varying faults generated by USV actuators

$$\hat{\omega}(t) = q(t) + K_\omega \tau(t) \tag{16}$$

$$\dot{q}(t) = -K_\omega \hat{\omega}(t) + K_\omega[A_{tr} u(t) - A_{tr} u_c(t)] \tag{17}$$

where, $\hat{\omega}(t) \in \mathbb{R}^3$ is the fault estimate, $q(t) \in \mathbb{R}^3$ denotes the auxiliary middle vector and $K_\omega \in \mathbb{R}^{3\times 3}$ denotes the design matrix.

On the basis of (5) and (16)-(17), we can obtain

$$\dot{\hat{\omega}}(t) = \dot{q}(t) + K_\omega \dot{\tau}(t)$$
$$= -K_\omega \hat{\omega}(t) + K_\omega[A_{tr} u(t) - A_{tr} u_c(t)]$$
$$+ K_\omega[-A_{tr} u(t) + A_{tr} u_c(t) + \omega(t)]$$
$$= K_\omega[\omega(t) - \hat{\omega}(t)]$$
$$= K_\omega \tilde{\omega}(t) \tag{18}$$

From (18), we can obtain

$$\dot{\tilde{\omega}}(t) = \dot{\omega}(t) - \dot{\hat{\omega}}(t)$$
$$= \dot{\omega}(t) - K_\omega \tilde{\omega}(t) \tag{19}$$

Select the Lyapunov function candidate as follows

$$V_\omega = \frac{1}{2}\tilde{\omega}^{\mathrm{T}}\tilde{\omega} \tag{20}$$

According to the derivation of Lyapunov function (19)-(20), Young's inequality, it has

$$\dot{V}_\omega = -\tilde{\omega}^{\mathrm{T}} K_\omega \tilde{\omega} + \tilde{\omega}^{\mathrm{T}}\dot{\omega}$$
$$\leq -\tilde{\omega}^{\mathrm{T}} K_\omega \tilde{\omega} + \frac{1}{2}\tilde{\omega}^{\mathrm{T}}\tilde{\omega} + \frac{1}{2}\omega_d^2$$
$$\leq -\lambda_{\min}\Big(K_\omega - \frac{1}{2}I_{3\times 3}\Big)\|\tilde{\omega}\|^2 + \frac{1}{2}\omega_d^2 \tag{21}$$

### C. Adaptive Tracking Control Design

The trajectory tracking control law for USVs under external time-varying disturbances, actuator faults, and input saturation is designed in three steps using a backstepping technique.

$$z_1 = \eta - \eta_d - \chi_1 \tag{22}$$
$$z_2 = \nu - \alpha_1 - \chi_2 \tag{23}$$
$$z_3 = u - \alpha_2 - \chi_3 \tag{24}$$

where $\alpha_i \in \mathbb{R}^3 (i = 1, 2)$ are the intermediate control vector to be designed later and $\chi_i \in \mathbb{R}^3 (i = 1, 2, 3)$ are produced by the actuator saturation compensator as follows:

$$\dot{\chi}_1 = -K_1 \chi_1 + J(\psi)\chi_2 \tag{25}$$
$$\dot{\chi}_2 = -M^{-1} K_2 \chi_2 + M^{-1}\chi_3 \tag{26}$$
$$\dot{\chi}_3 = -A_{tr} K_3 \chi_3 + A_{tr}\Delta u \tag{27}$$

where $K_i \in \mathbb{R}^3 (i = 1, 2, 3)$ are positive-definite design matrices.

**Step 1.** Along with (1), (22) and (25), it can be given that

$$\dot{z}_1 = J(\psi)\nu - \dot{\eta}_d + K_1 \chi_1 - J(\psi)\chi_2 \tag{28}$$

Select a Lyapunov function candidate

$$V_1 = \frac{1}{2}z_1^{\mathrm{T}} z_1 \tag{29}$$

On the basis of (28) and (29), we can obtain

$$\dot{V}_1 = z_1^{\mathrm{T}}[J(\psi)(z_2 + \alpha_1 + \chi_2) - \dot{\eta}_d + K_1 \chi_1 - J(\psi)\chi_2] \tag{30}$$

Design the first virtual intermediate vector $\alpha_1$ as:

$$\alpha_1 = -J^{\mathrm{T}}(\psi)(K_1 z_1 + K_1 \chi_1 - \dot{\eta}_d) \tag{31}$$

On the basis of (23), (28) and (31), we can obtain

$$\dot{z}_1 = J(\psi)(z_2 + \alpha_1) - \dot{\eta}_d$$
$$= -K_1 z_1 + J(\psi)z_2 \tag{32}$$

By utilizing (30) and (32), it can be given that

$$\dot{V}_1 = -z_1^{\mathrm{T}} K_1 z_1 + z_1^{\mathrm{T}} J(\psi) z_2 \tag{33}$$

**Step 2.** On the basis of (3), (23), we can obtain

$$\dot{z}_2 = M^{-1}[u(t) + d(t) - M\dot{\alpha}_1$$
$$- C(\nu)\nu - D(\nu)\nu + K_2\chi_2 - \chi_3] \tag{34}$$

Construct the Lyapunov function candidate $V_2$ as follows:

$$V_2 = V_1 + \frac{1}{2} z_2^{\mathrm{T}} M z_2 \tag{35}$$

Design the second virtual intermediate vector $\alpha_2$ as

$$\alpha_2 = -J^{\mathrm{T}}(\psi)z_1 - K_2(z_2 + \chi_2)$$
$$+ M\dot{\alpha}_1 + C(\nu)\nu + D(\nu)\nu - \hat{d}(t) \tag{36}$$

On the basis of (24), (34) and (36), we can obtain

$$\dot{z}_2 = M^{-1}[z_3 + \alpha_2 + \chi_3 + d(t)$$
$$- M\dot{\alpha}_1 - C(\nu)\nu - D(\nu)\nu + K_2\chi_2 - \chi_3]$$
$$= M^{-1}[z_3 - J^{\mathrm{T}}(\psi)z_1 - K_2 z_2 + \tilde{d}(t)] \tag{37}$$

On the basis of (33), (35) and (37), we can obtain

$$\dot{V}_2 = \dot{V}_1 + z_2^{\mathrm{T}} M \dot{z}_2$$
$$= -z_1^{\mathrm{T}} K_1 z_1 - z_2^{\mathrm{T}} K_2 z_2 + z_2^{\mathrm{T}} z_3 + z_2^{\mathrm{T}} \tilde{d}(t) \tag{38}$$

**Step 3.** The derivative of (24) along with (6) is

$$\dot{z}_3 = -A_{tr} u(t) + A_{tr} u_c(t) + \omega(t)$$
$$- \dot{\alpha}_2 + A_{tr} K_3\chi_3 - A_{tr}\Delta u \tag{39}$$

Construct the Lyapunov function candidate $V_3$ as follows:

$$V_3 = V_1 + \frac{1}{2} z_2^{\mathrm{T}} M z_2 + \frac{1}{2} z_3^{\mathrm{T}} A_{tr}^{-1} z_3 \tag{40}$$

Design the USV trajectory tracking control law as

$$\hat{u}_c(t) = -z_2 - K_3 z_3 - K_3\chi_3$$
$$+ u(t) + A_{tr}^{-1}\dot{\alpha}_2 - A_{tr}^{-1}\hat{\omega}(t) \tag{41}$$

On the basis of (39), (41) and $\Delta\tau = \tau_c(t) - \hat{\tau}_c(t)$, we can obtain

$$\dot{z}_3 = -A_{tr} u(t) + A_{tr} u_c(t) + \omega(t)$$
$$- \dot{\alpha}_2 + A_{tr} K_3\chi_3 - A_{tr}\Delta u$$
$$= -A_{tr} z_2 - A_{tr} K_3 z_3 + \tilde{\omega}(t) \tag{42}$$

On the basis of (38), (40) and (42), we can obtain

$$\dot{V}_3 = -z_1^{\mathrm{T}} K_1 z_1 - z_2^{\mathrm{T}} K_2 z_2 - z_3^{\mathrm{T}} K_3 z_3$$
$$+ z_2^{\mathrm{T}} \tilde{d}(t) + z_3^{\mathrm{T}} A_{tr}^{-1}\tilde{\omega}(t) \tag{43}$$

## D. Stability Analysis

*Theorem 1:* Consider the trajectory tracking control problem described by the three-degree-of-freedom USV motion mathematical model (1)-(7). Under Assumptions 1-2, the trajectory tracking control law (41) designed based on the combination of observer (8)-(9), (16)-(17) and backstepping method enables the USV to track the desired trajectory accurately. By adjusting the design parameters $K_d$, $K_\omega$, $K_1$, $K_2$, $K_3$, $\lambda_{\min}(K_1) > 0$, $\lambda_{\min}(K_2) > \frac{1}{2}$, $\lambda_{\min}(K_3) > 1$, $\lambda_{\min}(Q) > 2$, $\lambda_{\min}\left(K_\omega - \frac{1}{2}(A_{tr}^{-1})^{\mathrm{T}} A_{tr}^{-1}\right) > \frac{1}{2}$, ensures high-precision control of the closed-loop system.

**Proof**: Construct the Lyapunov function candidate $V$ as follows:

$$V = V_3 + V_d + V_\omega \tag{44}$$

On the basis of (15), (21) with (43) and (44), we can obtain

$$\dot{V} = \dot{V}_3 + \dot{V}_d + \dot{V}_\omega$$
$$= -z_1^{\mathrm{T}} K_1 z_1 - z_2^{\mathrm{T}} K_2 z_2 - z_3^{\mathrm{T}} K_3 z_3 + z_2^{\mathrm{T}} \tilde{d}(t)$$
$$+ z_3^{\mathrm{T}} A_{tr}^{-1}\tilde{\omega}(t) - \left[\frac{1}{2}\lambda_{\min}(Q) - \frac{1}{2}\right]||\tilde{d}||^2 + \frac{1}{2}||P\Psi||^2 c_d^2$$
$$- \lambda_{\min}\left(K_\omega - \frac{1}{2}I_{3\times3}\right)||\tilde{\omega}||^2 + \frac{1}{2}\omega_d^2$$
$$\leq -z_2^{\mathrm{T}}\left(K_2 - \frac{1}{2}I_{3\times3}\right)z_2 - z_3^{\mathrm{T}}\left(K_3 - \frac{1}{2}I_{3\times3}\right)z_3$$
$$- z_1^{\mathrm{T}} K_1 z_1 - \left[\lambda_{\min}\left(K_\omega - \frac{1}{2}A_{tr}^{-\mathrm{T}} A_{tr}^{-1}\right) - \frac{1}{2}\right]||\tilde{\omega}||^2$$
$$- \left[\frac{1}{2}\lambda_{\min}(Q) - 1\right]||\tilde{d}||^2 + \frac{1}{2}||P\Psi||^2 c_d^2 + \frac{1}{2}\omega_d^2$$
$$\leq \alpha V + \beta \tag{45}$$

Herein,

$$\alpha = \min\left\{\lambda_{\min}(K_1), \lambda_{\min}(K_2 - \frac{1}{2}I_{3\times3}),\right.$$
$$\lambda_{\min}(K_3 - I_{3\times3}), \frac{1}{2}\lambda_{\min}(Q) - 1,$$
$$\left.\lambda_{\min}\left(K_\omega - \frac{1}{2}(A_{tr}^{-1})^{\mathrm{T}} A_{tr}^{-1}\right) - \frac{1}{2}\right\} \tag{46}$$

$$\beta = \frac{1}{2}||P\Psi||^2 c_d^2 + \frac{1}{2}\omega_d^2 \tag{47}$$

Therefore, the design parameters should satisfy the conditions

$$\lambda_{\min}(K_1) > 0 \tag{48}$$

$$\lambda_{\min}(K_2) > \frac{1}{2} \tag{49}$$

$$\lambda_{\min}(K_3) > 1 \tag{50}$$

$$\lambda_{\min}(Q) > 2 \tag{51}$$

$$\lambda_{\min}\left(K_\omega - \frac{1}{2}(A_{tr}^{-1})^{\mathrm{T}} A_{tr}^{-1}\right) > \frac{1}{2} \tag{52}$$

Solving (45), we obtain

$$0 \leq V(t) \leq \frac{\beta}{\alpha} + \left[V(0) - \frac{\beta}{\alpha}\right]e^{-\alpha t} \tag{53}$$

The $V(t)$, $||z_1||$, $||z_2||$, $||z_3||$, $\tilde{d}(t)$, $\tilde{\omega}(t)$, $\dot{\tilde{d}}(t)$ and $\tilde{\omega}(t)$ are bounded, According to (53), it follows that

$$||\eta - \eta_d - \chi_1|| \leq \sqrt{\frac{2\beta}{\alpha} + 2\left[V(0) - \frac{\beta}{\alpha}\right]e^{-\alpha t}} \quad (54)$$

Construct the Lyapunov function candidate $V_\chi$ as follows:

$$V_\chi = \frac{1}{2}\chi_1^T\chi_1 + +\frac{1}{2}\chi_2^T M\chi_2 + \frac{1}{2}\chi_3^T A_{tr}^{-1}\chi_3 \quad (55)$$

On the basis of (25)-(27), $||J^{-1}(\psi)|| = 1$ and Young's inequality, it has

$$\begin{aligned}
\dot{V}_\chi = &-\chi_1^T K_1\chi_1 + \chi_1^T J(\psi)\chi_2 - \chi_2^T K_2\chi_2 \\
&+ \chi_2^T\chi_3 - \chi_3^T K_3\chi_3 + \chi_3^T\Delta u \\
\leq &-\chi_1^T\left(K_1 - \frac{1}{2}I_{3\times3}\right)\chi_1 - \chi_2^T\left(K_2 - I_{3\times3}\right)\chi_2 \\
&- \chi_3^T\left(K_2 - I_{3\times3}\right)\chi_3 + \frac{1}{2}||\Delta u||^2 \\
\leq &-2u_\chi V_\chi + C_\chi \quad (56)
\end{aligned}$$

Herein,

$$\begin{aligned}
u_\chi = &\min\left\{\lambda_{\min}(K_1\frac{1}{2}I_{3\times3}), \lambda_{\min}(K_2 - I_{3\times3})M_{-1},\right. \\
&\left.\lambda_{\min}(K_3 - I_{3\times3})A_{tr}\right\} \quad (57)
\end{aligned}$$

$$C_\chi = \frac{1}{2}sup_{t\geq0}||\Delta\tau||^2 \quad (58)$$

with $\lambda_{\min}(K_1) > \frac{1}{2}$, $\lambda_{\min}(K_2) > 1$ and $\lambda_{\min}(K_3) > 1$. It is obtained from (56) that

$$0 \leq ||\chi_1|| \leq \sqrt{\frac{C_\chi}{u_\chi} + 2\left[V_\chi(0) - \frac{C_\chi}{2u_\chi}\right]e^{-2u_\chi t}} \quad (59)$$

Then,

$$\Omega_\eta = \left\{||\eta - \eta_d|| \leq \delta_\eta, \delta_\eta > \sqrt{\frac{2\beta}{\alpha}} + \sqrt{\frac{C_\chi}{u_\chi}}\right\} \quad (60)$$

From the definitions of the symbols $\alpha$, $u_\chi$, $\beta$ and $C_\chi$, the tracking error of the USV can be made small by adjusting the design parameter $K_d$, $K_\omega$, $K_1$, $K_2$, $K_3$, $\lambda_{\min}(K_1) > 0$, $\lambda_{\min}(K_2) > \frac{1}{2}$, $\lambda_{\min}(K_3) > 1$, $\lambda_{\min}(Q) > 2$, $\lambda_{\min}\left(K_\omega - \frac{1}{2}(A_{tr}^{-1})^T A_{tr}^{-1}\right) > \frac{1}{2}$. Thus, the USV accurately tracks the desired trajectory, the correctness of Theorem 1 is verified.

## IV. ILLUSTRATIVE RESEARCH

The robustness and effectiveness of the designed USV trajectory tracking control strategy is verified on the full scale 'MS The Waterfall' in the following.

$$M = \begin{bmatrix} m_{11} & 0 & 0 \\ 0 & m_{22} & m_{23} \\ 0 & m_{32} & m_{33} \end{bmatrix} \quad (61)$$

Here,

$$m_{11} = 2.7521 \times 10^5 \quad (62)$$
$$m_{22} = 7.6348 \times 10^5 \quad (63)$$
$$m_{23} = 7.3803 \times 10^5 \quad (64)$$
$$m_{32} = -7.3803 \times 10^5 \quad (65)$$
$$m_{33} = 6.690963 \times 10^7 \quad (66)$$

The $C(\nu)$ matrix as follows:

$$C(\nu) = \begin{bmatrix} 0 & 0 & c_{13}(\nu) \\ 0 & 0 & c_{23}(\nu) \\ c_{31}(\nu) & c_{32}(\nu) & 0 \end{bmatrix} \quad (67)$$

Here,

$$c_{31}(\nu) = 7.6348 \times 10^5 v - 7.3803 \times 10^5 r \quad (68)$$
$$c_{32}(\nu) = -2.7521 \times 10^5 u \quad (69)$$
$$c_{13}(\nu) = -7.6348 \times 10^5 v + 7.3803 \times 10^5 r \quad (70)$$
$$c_{23}(\nu) = 2.7521 \times 10^5 u \quad (71)$$

The damping matrix are given as follow:

$$D(\nu) = \begin{bmatrix} d_{11}(\nu) & 0 & 0 \\ 0 & d_{22}(\nu) & d_{23}(\nu) \\ 0 & d_{32}(\nu) & d_{33}(\nu) \end{bmatrix} \quad (72)$$

Here,

$$d_{11}(\nu) = 2.5 \times 10 + 2.375 \times 10^3|u| \quad (73)$$
$$\begin{aligned}d_{22}(\nu) = &9.865 \times 10^3 + 8.843 \times 10^3|u| \\ &+ 2.83 \times 10^3|v| + 9.537 \times 10^5|r| \quad (74)\end{aligned}$$
$$d_{23}(\nu) = 1.375 \times 10^3 + 1.3495 \times 10^5|v \quad (75)$$
$$d_{32}(\nu) = 7.11 \times 10^2 + 1.11774 \times 10^5|u| \quad (76)$$
$$\begin{aligned}d_{33}(\nu) = &2.813355 \times 10^6 + 6.047978 \times 10^6|u| \\ &+ 1.18865 \times 10^7|v| + 1.56288 \times 10^8|r| \quad (77)\end{aligned}$$

Set the reference trajectory of the USV to be ( [24])

$$y_d = 140 + 30\sin(0.01x_d)\,\text{m}. \quad (78)$$

where, $x_d = t$, $y_d = 140 + 30\sin(0.01t)$, and $\psi_d = \arctan(y_d/x_d)$. Set the initial states as $\eta(0) = \left[20\text{m}, 100\text{m}, \frac{10\pi}{180}\text{rad}\right]^T$ and $\nu(0) = [0\text{m/s}, 0\text{m/s}, 0\text{rad/s}]^T$. The USV is also susceptible to noise disturbances while navigation, defined as follows noise disturbances as ( [25]):

$$x_w = \frac{4\lambda_0\omega_0\sigma_0 s}{s^2 + 2\lambda_0\omega_0 s + \omega_0^2}n \quad (79)$$

$$y_w = \frac{4\lambda_0\omega_0\sigma_0 s}{s^2 + 2\lambda_0\omega_0 s + \omega_0^2}n \quad (80)$$

$$\psi_w = \frac{6\lambda_0\omega_0\sigma_0 s}{s^2 + 2\lambda_0\omega_0 s + \omega_0^2}n * (\pi/180) \quad (81)$$

where $\lambda_0 = 0.1$, $\omega_0 = 0.8$, $\sigma_0 = 0.5$, and $n$, $x_m$, $y_m$, and $\psi_m$ are noise disturbances. Thus, $x + x_w + x_m$, $y + y_w + y_m$, and $\psi + \psi_w + \psi_m$. The formula for the connection filter is

$$f_{lp} \times f_{no} = \frac{1}{t_f s + 1} \times \frac{s^2 + 2\eta_n\omega_n s + \omega_9}{s^2 + 2\omega_n s + \omega_9}, \quad (82)$$

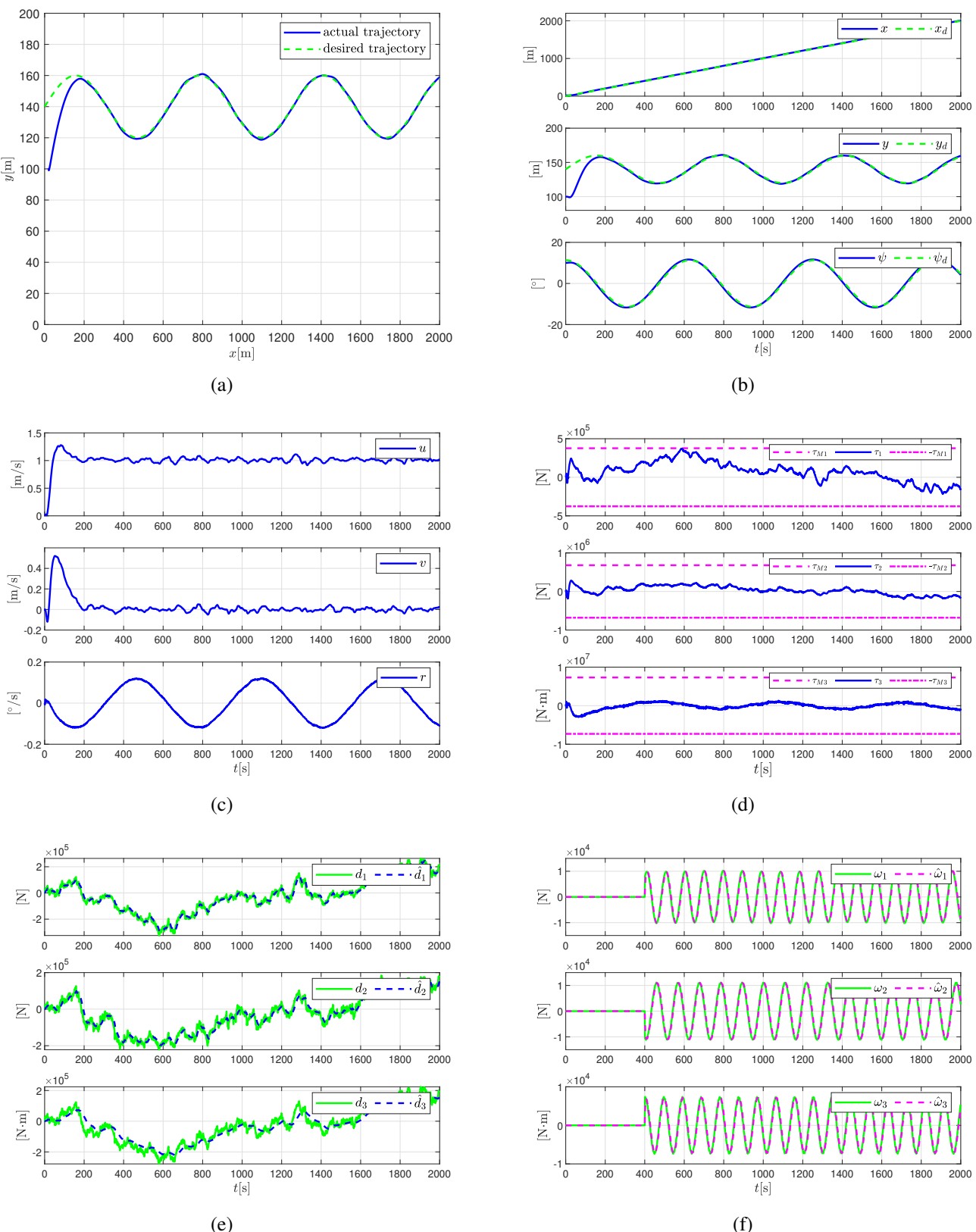

Fig. 1: Illustrative results. (a) The track diagram of the in the xy-plane. (b) Actual and desired position of the USV. (c) Velocities in surge, sway and yaw (d) Control input signals. (e) Actual disturbances $d_1, d_2, d_3$ and estimated values $\hat{d}_1, \hat{d}_2, \hat{d}_3$. (f) Actual faults $\omega_1, \omega_2, \omega_3$ and estimated values $\hat{\omega}_1, \hat{\omega}_2, \hat{\omega}_3$.

where $t_f = 10$, $\eta_n = 0.1$, and $\omega_n = 0.83$.

The external time-varying disturbances are set as

$$T = diag(1000, 1000, 1000) \tag{83}$$
$$\Psi = 2 \times diag(15000, 15000, 15000) \tag{84}$$

The time-varying fault parameters are as follows:
$\omega(t) = 2 \times$

$$\begin{cases} [0,0,0]^{\mathrm{T}}, 0 \leq t \leq 400 \\ \begin{bmatrix} 100\sin(0.01t) + 5000\sin(0.065t) \\ 100\cos(t - \dfrac{\pi}{6}) + 5500\sin(0.058t) \\ 100\cos(t) + 3600\sin(0.0664t) \end{bmatrix}, t > 400 \end{cases} \tag{85}$$

We set the design parameter matrices as $K_d = diag([8 \times 10^5, 8 \times 10^5, 1 \times 10^8])$, $K_\omega = diag([50, 50, 50])$, $K_1 = diag([0.02, 0.02, 0.03])$, $K_2 = diag([6 \times 10^5, 1.1 \times 10^6, 2 \times 10^9])$ and $K_3 = diag([1.2, 1.2, 1.2])$.

The simulation results are plotted in Figures 1(a), 1(b), 1(c), 1(d), 1(e) and 1(f). Figure 1(a) gives that the designed control strategy enables the USV to track the reference trajectory with small error. Figure 1(b) gives the time curves of the actual position of the USV's $(x, y)$ and the bow sway angle $\psi$. Figure 1(c) shows that the surge velocity, sway velocity and yaw rate are bounded and reasonable. Figure 1(d) illustrates that the control input signals in surge $\tau_1$, in sway $\tau_2$ and in yaw $\tau_3$ are subject to the saturation. The time curves for actual and estimated disturbances are given in Figure 1(e), and the time curves for actual and estimated faults are given in Figure 1(f).

In summary, Figures 1(a)-1(f) shows that the proposed control strategy exhibits the practical trajectory tracking control performance in the simultaneous presence of disturbances and faults under saturation. These simulation results verify the validity of the control method under time-varying disturbance and faults.

## V. CONCLUSION

This paper has addressed the problem of separate observer-based estimation and control for unmanned autonomous vehicles with disturbances and faults under input saturation. The disturbance observer on-line estimates and compensates for time-varying disturbances and the actuator fault observer on-line estimates and compensates for actuator faults separately. An auxiliary dynamic filter is used to attenuate the adverse effects of input saturation. The stability analysis of the control system is carried out by means of Lyapunov stability theory, which has shown that all signals of the closed-loop system are globally uniform and ultimately bounded. We have extended the results presented herein to the dynamic event-triggered case in the reference [26].

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
