# OpenReview forum: "Separate Observer-based Estimation and Control for Unmanned Autonomous Vehicles with Disturbances and Faults under Input Saturation"
_IEEE.org/ICIST/2024/Conference — IEEE ICIST 2024 Conference Submission_

### Official Review · Reviewer_2WTd · 2024-08-24
**The quality of the article is good.**

**Rating:** 8
**Confidence:** 4

**Review:**

This paper discusses the separate observer-based estimation and control for unmanned autonomous vehicles with disturbances and faults under input saturation effects. A trajectory tracking control scheme is designed with integrating the separate observers. The disturbance observer on-line estimates and compensates for time-varying uncertain disturbances. The actuator fault observer enables on-line estimation and compensation of actuator faults. An auxiliary dynamic filter is used to attenuate the adverse effects of control saturation. The Lyapunov stability theory demonstrates that all the signals of the closed-loop system are global uniformly ultimately bounded. Illustrations show the effectiveness of this scheme.

---

### Official Review · Reviewer_5nJW · 2024-08-30
**Accept**

**Rating:** 10
**Confidence:** 5

**Review:**

1.	Some control methods and theories are introduced in this paper. If these methods are not proposed by the author for the first time, it is recommended to supplement the corresponding references.
2.	The paper reviews the existing research results in detail in the introduction but lacks an overview of the main innovation points of this paper. It is recommended to add a paragraph at the end of the introduction that summarizes the contribution of the paper.
3.	To make it easier for the reader to understand, it is recommended to add a flow chart or block diagram to the simulation results section.

---

### Official Review · Reviewer_Juj7 · 2024-09-02
**This paper can be accepted.**

**Rating:** 6
**Confidence:** 5

**Review:**

This paper discusses the separate observer-based estimation and control for unmanned autonomous vehicles with disturbances and faults under input saturation effects.
The reviewer's comments are as follows:
1.The English grammar and format of this manuscript could be further polished and checked carefully.
2.The format of the references is not uniform.
3.To enhance the quality of the manuscript, it is recommened refining the language to improve readability and ensuring that the concepts are communicated accurately.
4.In addition to the contribution of the proposed work, it is suggested to discuss its main limitations/shortcomings.
5.Please add a structural flowchart of the paper's framework to enhance the readability of the paper.

---

### Decision · Program_Chairs · 2024-09-06

Accept (Oral)